

# Brain mechanism of unfamiliar and familiar voice processing: an activation likelihood estimation meta-analysis

YuXiang Sun, Lili Ming, Jiamin Sun, FeiFei Guo, Qiufeng Li and Xueping Hu

[1] School of Linguistic Science and Art, Jiangsu Normal University, Xuzhou, Jiangsu, China
[2] Key Laboratory of Language and Cognitive Neuroscience of Jiangsu Province, Collaborative Innovation Center for Language Ability, Xuzhou, China

Corresponding author
Xueping Hu, huxpxp@163.com

## ABSTRACT

Interpersonal communication through vocal information is very important for human society. During verbal interactions, our vocal cord vibrations convey important information regarding voice identity, which allows us to decide how to respond to speakers (*e.g.*, neither greeting a stranger too warmly or speaking too coldly to a friend). Numerous neural studies have shown that identifying familiar and unfamiliar voices may rely on different neural bases. However, the mechanism underlying voice identification of individuals of varying familiarity has not been determined due to vague definitions, confusion of terms, and differences in task design. To address this issue, the present study first categorized three kinds of voice identity processing (perception, recognition and identification) from speakers with different degrees of familiarity. We defined voice identity perception as passively listening to a voice or determining if the voice was human, voice identity recognition as determining if the sound heard was acoustically familiar, and voice identity identification as ascertaining whether a voice is associated with a name or face. Of these, voice identity perception involves processing unfamiliar voices, and voice identity recognition and identification involves processing familiar voices. According to these three definitions, we performed activation likelihood estimation (ALE) on 32 studies and revealed different brain mechanisms underlying processing of unfamiliar and familiar voice identities. The results were as follows: (1) familiar voice recognition/identification was supported by a network involving most regions in the temporal lobe, some regions in the frontal lobe, subcortical structures and regions around the marginal lobes; (2) the bilateral superior temporal gyrus was recruited for voice identity perception of an unfamiliar voice; (3) voice identity recognition/identification of familiar voices was more likely to activate the right frontal lobe than voice identity perception of unfamiliar voices, while voice identity perception of an unfamiliar voice was more likely to activate the bilateral temporal lobe and left frontal lobe; and (4) the bilateral superior temporal gyrus served as a shared neural basis of unfamiliar voice identity perception and familiar voice identity recognition/identification. In general, the results of the current study address gaps in the literature, provide clear definitions of concepts, and indicate brain mechanisms for subsequent investigations.

## INTRODUCTION

Interpersonal communication through acoustic information is a very important and efficient method for human beings and society (*Scott, 2019*). In the process of communication, the acoustic signals generated by the vibration of our vocal cords simultaneously convey two important information streams: linguistic information and paralinguistic information (*Mathiak et al., 2007*; *Relander & Rama, 2009*; *Zhang et al., 2016*). Language information (what) allows us to understand and respond to what the speaker is saying; paralinguistic information, such as voice identity (who), allows us to ascertain and know the identity of the speaker (*Belin et al., 2011*; *Belin, Fecteau & Bedard, 2004*; *Kuhl, 2011*; *Lattner, Meyer & Friederici, 2005*; *Von Kriegstein et al., 2005*) and further to decide how to respond. For example, when we receive a call, we can immediately judge whether it is a call from an acquaintance or a stranger through voice identity (*Kuhl, 2011*) to determine our communication mode. Therefore, the familiarity of voice identity is self-evident for interpersonal communication, for instance, we do not use very intimate words with a stranger, nor do we maintain an excessive sense of distance with a familiar person.

*Belin et al. (2000)* first discovered voice-selective sensitive regions in the human brain, that is, regions located in bilateral superior temporal sulcus/gyrus (STS/STG) selectively sensitive to human voices. Since then, the neural basis of voice perception has been explored and confirmed, and the number of studies on voice identity processing have gradually increased. However, the existing research has some shortcomings. For example, except for some studies on unfamiliar voice identity processing, most of the current studies focus on familiar voice identity processing. To date, research has determined the rough outline of selective processing of unfamiliar human voices, indicating that the bilateral STS/STG region is recruited (*Belin et al., 2000*; *Fecteau et al., 2004*). Moreover, previous studies have shown that extratemporal extended regions (*e.g.*, the inferior frontal gyrus, IFG), except for the right superior temporal gyrus/sulcus and the middle temporal gyrus (rSTG/S and MTG), are preferentially activated by repeatedly presented voices (*Belin & Zatorre, 2003*; *Bonte et al., 2014*; *Joassin, Maurage & Campanella, 2011*; *Von Kriegstein et al., 2003*) or familiar voices (*Andics, McQueen & Petersson, 2013*; *Hasan et al., 2016*; *Hoelig et al., 2017*; *Von Kriegstein & Giraud, 2004*). One conclusion can be drawn from the literature: the voice identity of the speaker is processed in a specific area of the brain, voice identities that differ in familiarity may be processed in different areas (*Maguinness, Roswandowitz & Von Kriegstein, 2018*). However, there are still some problems to be improved upon in recent research, such as unclear and confusing terms used in voice identity processing with different levels of familiarity and the differences in experimental task design. Therefore, the existing research is effectively unable to determine the mechanism by which listeners distinguish voice identities among speakers with varying familiarity.

For the mixed use of terms in voice identity with different degrees of familiarity, the present study divided voice identity into three levels based on the characteristics of different stages of voice identity processing: "voice identity perception", "voice identity recognition" and "voice identity identification" (Table 1). Combined with the existing research in the
**Table 1  Three levels of voice identity processing.**

|  | Voice identity perception | Voice identity recognition | Voice identity identification |
|---|---|---|---|
| Experimental materials | Unfamiliar | Familiar (Acoustically) | Familiar (Multimodal) |
| Task design | Passively listen to a voice or determine if the voice is human | Determine if the sound you hear is acoustically familiar | Ascertain whether you can associate a voice with a name or a face |

field of voice identity processing, this paper provides a simple summary according to the terms used in fMRI research on voice identity processing, the nature of experimental materials (familiar or not) and the perspective of task design.

First, voice identity perception (VIP) occurs when processing speech by unfamiliar speakers; the listener performs low-order acoustic discrimination for the input stimulus, then perceives and judges whether it was a human voice. Accordingly, there are two cases included in VIP: the vocal stimuli used are completely unfamiliar to the subject (*Belin et al., 2000*; *Warren et al., 2006*), which means that there is no recognition of the speaker's identity, and subjects only listened to the experimental stimulus one time, and most of the tasks were passive listening (*Agus et al., 2017*; *Latinus et al., 2013*; *Pernet et al., 2015*). Even when active tasks were involved, subjects were only asked to press buttons to determine whether the stimulus they heard was the sound produced by the vibration of human vocal cords (*Lee et al., 2015*; *Roswandowitz, Swanborough & Fruehholz, 2021*). Most of these studies are related to voice selectivity and species specificity (*Fecteau et al., 2004*; *Fecteau et al., 2005*). Therefore, previous studies in this area have repeatedly confirmed the existence of corresponding voice-sensitive areas in the human brain, and a corresponding 'voice localizer' has also been proposed (*Pernet et al., 2015*), which is often used in various voice studies to locate the brain regions sensitive to human voice and to conduct in-depth brain mechanism analysis. Specifically, there are many extensions of the research on voice perception, such as voice emotion perception and voice gender perception, which leads to a broad range of research perspectives in the field of voice perception and further leads to the research on VIP mixed with the research on nonvoice identity processing.

Second, voice identity recognition (VIR) occurs when processing an acoustically familiar voice identity, and the subjects perform acoustic matching recognition of the input stimulus. Therefore, there are two cases included in VIR: before the formal experiment, the listener learns/has been exposed to the voice stimuli that would be used in the subsequent experiment through auditory unimodal learning (*Belin & Zatorre, 2003*; *Von Kriegstein et al., 2003*). Although the learning time could be short or long, the listener could still remember the stimulus at the acoustic memory level during the task. Most of the tasks require the listener to judge whether the voice stimuli presented in the current trial have been learned/presented before, to determine whether the voice is being repeatedly presented, or to detect whether the voice changes. In other words, experimental designs of VIR, such as learning recognition tasks (*Andics et al., 2010*; *Zaeske, Hasan & Belin, 2017*),

 

delayed matching tasks (*Bonte et al., 2014*; *Rama & Courtney, 2005*; *Rama et al., 2004*; *Relander & Rama, 2009*), n-back tasks (*Mathiak et al., 2007*; *Stevens, 2004*), and change detection tasks (*Zhang et al., 2016*), are quite diverse. It should be noted that the familiarity of voice in those studies of VIR described in the current paper does not reach the level of voice identity identification (VII) and only pertains to acoustic memory or acoustic familiarity of voice identity.

Finally, VII occurs when processing the voice of a familiar speaker in daily life; listeners can generate associations related to faces, relationships, social status, and so, on in response to the input stimuli. Hence, there are two situations included in VII: listeners are completely familiar with the vocal stimuli (in daily life) used in the experiment (*Birkett et al., 2007*; *Von Kriegstein & Giraud, 2004*; *Von Kriegstein et al., 2005*). In other words, listeners have multimodal semantic information, also known as biographical information, about the voice stimuli stored in their memory (*Mathias & Von Kriegstein, 2014*). Factually, familiarity can also be achieved by setting up training sessions in which subjects learn faces or names corresponding to voice stimuli (*Andics, McQueen & Petersson, 2013*; *Joassin, Maurage & Campanella, 2011*; *Latinus, Crabbe & Belin, 2011*) to generate multimodal semantic information associations about the speaker. Most of the experimental tasks were designed for the subjects to passively listen to those familiar voices (*Ogg et al., 2019*; *Shah et al., 2001*) or to judge whether the voice stimuli presented in the current trial were familiar or not (*Aglieri et al., 2021*; *Hasan et al., 2016*; *Hoelig et al., 2017*). The familiarity of VII is different from the acoustic familiarity of VIR, which refers to the multimodal semantic information about the face and name of the voice stimulus, or the voice identity is completely familiar to the subject in daily life.

In this meta-analysis, we define VIR/I as familiar voice processing and VIP as unfamiliar voice processing; we and compare unfamiliar voice processing with familiar voice processing. In contrast to studies on VIP, the literature on VIR and VII with familiar speakers is extensive. The available evidence suggests that there may be differences between brain mechanisms for recognition/identification of familiar voices and brain mechanisms for perception of unfamiliar voices. Importantly, although this difference has received some attention in neuroimaging studies of voice processing, due to the diverse task design, the intermixed use of terms and unclear concept definition in different voice identity studies, the description of brain mechanisms of voice identity with different degrees of familiarity needs to be improved. Hence, based on the different clearly defined levels of voice perception, "VIP" and "VIR/I", the present study adopted the meta-analysis method of activation likelihood estimation (ALE) to summarize the similarities and differences in brain mechanisms between unfamiliar voice identity and familiar voice identity processing. In summary, the current study could provide a clear definition of these concepts and a reference for brain mechanism patterns for subsequent research on voice identity processing.

## MATERIALS AND METHODS

### Literature search and selection

In view of the wide range of aspects in the field of voice research, including both voice "identity research" and "voice emotion research", the keywords chosen for this meta-analysis needed to target "VIP" relatively accurately. Considering that the research on "VIP" is often mixed with that on "voice emotion perception", to avoid the potential influence of "emotion perception" on the coordinate "VIP" data, "perception" was omitted from the keywords after comprehensive consideration. Instead, "recognition" and "identification", two relatively high-level identity processing terms, were selected to specify the research on "VIP" as much as possible. In addition, as mentioned in the Introduction section, the research on "VIP" has mixed terms, so our search results still contained quite a few studies on VIP.

It should be noted all the literature obtained by the search was screened according to VIP as defined in the introduction. In other words, the VIP studies in this study were all focused on the selective sensitivity of the human voice, that is, the lowest level of acoustic signal based on the perception of stranger identity. Studies that did not include voice identity processing, such as voice emotion processing, were excluded.

YuXiang Sun and XuePing Hu performed the search and identified relevant manuscripts, with no disagreements regarding manuscript eligibility. The literature search and selection were independently completed by the first author. Data extraction was independently completed by the first author. In the Web of Science database (voice recognition * OR voice identification * OR speaker recognition * OR speaker identification * OR talker recognition * OR talker identification *) AND (fMRI) were keywords searched for studies published before December 15, 2021, and a total of 378 studies were obtained. Through the advanced screening provided by Web of Science, 314 remaining papers were filtered after review papers, clinical trials, meeting, abstracts, case reports, unspecified, books and data papers were excluded.

According to the criterion of reading the title first and then the abstract, the literature was divided into three categories: VIP, VIR and VII. The screening criteria were as follows: (1) Exclusion of studies on VIP, VIR and VII that were not defined in the introduction; (2) exclusion of speech studies; (3) exclusion of studies of nonauditory modalities; (4) exclusion of non-fMRI studies; (5) exclusion of studies based on ROI analysis or without reported coordinates; (6) exclusion of studies with abnormal populations; and (7) exclusion of studies with nonadult subjects.

A total of 17 available studies were obtained by screening, including five studies on VIP, seven studies on VIR, and five studies on VII. Although a total of 17 articles met the recommended minimum requirements for ALE analysis (*Eickhoff et al., 2016*) after exhaustively using as many keywords as possible for voice identity processing, a meta-analysis with a small sample size may not have had sufficient statistical power. Therefore, two schemes were carried out to complement the literature: the data of voice identity processing in multimodal identity processing that appeared in 314 studies were considered, and a total of five studies were obtained, including one study on VIP, one study on VIR,
and three studies on VII. According to the previous practice of supplementing the literature (*Parola et al., 2020*), the studies that could be included in the meta-analysis were screened from the references of 17 eligible studies. This screening process ensured that all papers appeared in the Web of Science database and were voice identity processing studies. A total of 10 articles were obtained, including four studies on VIP, three studies on VIR, and 3 studies on VII. After careful reading, it was found that the keywords used in these studies did not have relevant pronouns related to voice identity processing but used other keywords to refer to them, such as "auditory perception" and "familiarity".

In summary, a total of 32 articles were obtained by the retrieval and screening procedure, including 10 articles on VIP, 11 articles on VIR, and 11 articles on VII. A summary of the literature is shown in Table 2,  and the process of literature retrieval and selection is shown in Fig. 1.

## ALE meta-analysis

ALE is a coordinate-based meta-analysis method (CBMA) first proposed by *Turkeltaub et al. (2002)*. Since then, researchers have continuously updated and iterated the algorithm to obtain higher statistical power (*Eickhoff et al., 2012*; *Eickhoff et al., 2009*; *Turkeltaub et al., 2012*). The basic logic of the ALE method is to model the activation focus as a 3D Gaussian distribution and then calculate the activation distribution map of the meta-analysis with each experiment as the unit. Specifically, to estimate the probability of cross-experimental activation of each voxel under certain conditions, the ALE method first calculates the "modelled activation" (MA) map of each experiment separately: the MA map of each experiment was calculated by combining the probability that a single voxel is included in each peak activation point of the experiment in the unit of voxel. Then, we took the union of these individual MA maps and calculated the ALE value based on the MA value to obtain the ALE value across the experiments. Finally, the possibility of activation across experiments was tested for significance. The null hypothesis of the ALE method was that there is no coincidence between the MA maps of each experiment in the meta-analysis, and all coincidence is caused by random factors. The ALE results calculated by the actual coordinates were compared with the ALE values of the null hypothesis for the significance test to obtain the results of the meta-analysis. For more detailed statistical principles of the ALE algorithm, please refer to *Eickhoff et al. (2009)*, *Eickhoff et al. (2012)* and *Turkeltaub et al. (2012)*.

According to the requirements of the meta-analysis, the fMRI coordinate data of activation peak points for voice identity processing in all 32 studies were manually exported and sorted into text documents. Then, the coordinate conversion function of GingerALE software was used to convert all coordinates reported in MNI space into Talairach space coordinates. The converted coordinates were then exported. After the coordinates were exported, the coordinate data were arranged into the coordinate data format required by GingerALE software, and then the ALE method was used for coordinate-based meta-analysis.

Four different ALE analyses were performed using GingerALE 3.0.2 (*Eickhoff et al., 2009*): an overall analysis of all the included voice identity processing studies, and an
**Table 2   Summary of studies included in the meta-analysis.**

| Literature | Number of participants | Study category | Familiarity of voice | Task | Standard coordinate space |
|---|---|---|---|---|---|
| *Aglieri et al. (2021)* | 40 | Identification | Familiar with the name associated with the voice | Determine the identity (name) corresponding to the voice | MNI |
| *Agus et al. (2017)* | 22 | Perception | Strange | Passive listening | MNI |
| *Andics et al. (2010)* | 24 | Recognition | Familiar with the acoustic information of the voice | Ascertain whether the voice is consistent | MNI |
| *Andics, McQueen & Petersson (2013)* | 15 | Identification | Familiar with the name associated with the voice | Determine the identity (name) corresponding to the voice | MNI |
| *Belin et al. (2000)* | 6 | Perception | Strange | Passive listening | TAL |
| *Belin & Zatorre (2003)* | 14 | Recognition | Familiar with the acoustic information of the voice | Passive listening to repeated voice | TAL |
| *Birkett et al. (2007)* | 11 | Identification | Fully familiar with the voice identity | Identify familiar voices | TAL |
| *Bonte et al. (2014)* | 10 | Recognition | Familiar with the acoustic information of the voice | Ascertain whether the voice is consistent | TAL |
| *Fecteau et al. (2004)* | 15 | Perception | Strange | Passive listening | TAL |
| *Fecteau et al. (2005)* | 15 | Perception | Strange | Passive listening | TAL |
| *Hasan et al. (2016)* | 5 | Identification | Familiar with the face associated with the voice | Determine the identity (face) corresponding to the voice | MNI |
| *Hoelig et al. (2017)* | 18 | Identification | Familiar with the face associated with the voice | Determine the identity (face) corresponding to the voice | MNI |
| *Joassin, Maurage & Campanella (2011)* | 14 | Identification | Familiar with the face associated with the voice | Determine the identity (face) corresponding to the voice | MNI |
| *Latinus et al. (2013)* | 48 | Perception | Strange | Passive listening | MNI |
| *Latinus, Crabbe & Belin (2011)* | 16 | Identification | Familiar with the name associated with the voice | Determine the identity (name) corresponding to the voice | MNI |

**Table 2** (*continued*)

| Literature | Number of participants | Study category | Familiarity of voice | Task | Standard coordinate space |
|---|---|---|---|---|---|
| *Lee et al. (2015)* | 12 | Perception | Strange | Determine whether the voice is human | MNI |
| *Mathiak et al. (2007)* | 10 | Recognition | Familiar with the acoustic information of the voice | Determine whether the voice is repeated | MNI |
| *Ogg et al. (2019)* | 18 | Identification | Familiar with the name and face associated with the voice | Do not respond to voice stimulation, but listen passively | MNI |
| *Pernet et al. (2015)* | 218 | Perception | Strange | Passive listening | MNI |
| *Rama et al. (2004)* | 14 | Recognition | Familiar with the acoustic information of the voice | Determine whether the voice is repeated | TAL |
| *Rama & Courtney (2005)* | 12 | Recognition | Familiar with the acoustic information of the voice | Determine whether the voice is repeated | TAL |
| *Relander & Rama (2009)* | 10 | Recognition | Familiar with the acoustic information of the voice | Determine whether the voice is repeated | TAL |
| *Roswandowitz, Swanborough & Fruehholz (2021)* | 29 | Perception | Strange | Determine whether the voice is human | MNI |
| *Shah et al. (2001)* | 10 | Identification | Fully familiar with the voice identity | Do not respond to voice stimulation, but listen passively | TAL |
| *Stevens (2004)* | 9 | Recognition | Familiar with the acoustic information of the voice | Determine whether the voice is repeated | TAL |
| *Von Kriegstein et al. (2003)* | 14 | Recognition | Familiar with the acoustic information of the voice | Ascertain whether the voice is consistent | TAL |
| *Von Kriegstein & Giraud (2004)* | 9 | Identification | Fully familiar with the voice identity | Identify familiar voices | TAL |
| *Von Kriegstein et al. (2005)* | 9 | Identification | Fully familiar with the voice identity | Identify familiar voices | TAL |
| *Warren et al. (2006)* | 12 | Perception | Strange | Passive listening | MNI |
| *Watson et al. (2014)* | 40 | Perception | Strange | Passive listening | MNI |
| *Zaeske, Hasan & Belin (2017)* | 24 | Recognition | Familiar with the acoustic information of the voice | Determine whether the voice is repeated | MNI |
| *Zhang et al. (2016)* | 18 | Recognition | Familiar with the acoustic information of the voice | Ascertain whether the voice changes | TAL |

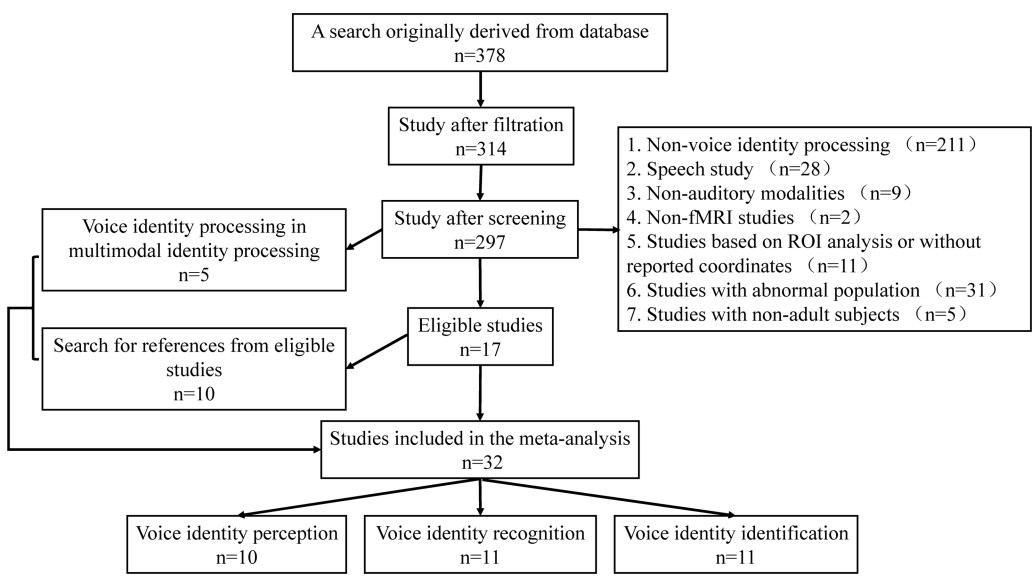

**Figure 1 Literature search and selection process diagram.**

analysis of the combination of VIR and VII research were conducted to explore the brain activation of "familiar" voice identity processing. To investigate the brain activation of "unfamiliar" vocal identity processing by analysing studies on VIP and to explore the brain activation of familiar voice identity *versus* unfamiliar voice identity, results from analyses and were combined and compared. In the current study, VIR and VII results were combined and compared to those of VIP because the former two are more familiar voice identity processing than the latter, and the number of meta-analysis literatures can reach 22 after the combination of the former two, which exceeds the recommended minimum number of studies required by ALE analysis. According to the number of studies in each analysis, cluster FWE (*Eickhoff et al., 2016*) was used in ALE analysis for and , and the cluster distribution threshold (cluster-level FWE) was set as $P < 0.01$. The cluster formation threshold was set to $P < 0.05$ and 1,000 permutations. For the ALE analysis, because the research on VIP only included 10 articles, voxel FWE was adopted according to (*Eickhoff et al., 2016*), and the threshold was set as an uncorrected threshold of $P < 0.05$ and 1,000 permutations. Referring to the recommendations of *Eickhoff et al. (2017)*, the ALE comparison and conjoint analysis of did not use FDR correction, but an uncorrected threshold of $p < 0.05$ and a minimum cluster standard of 200 mm3 were employed.

# ALE ANALYSIS RESULTS

## Voice identity processing

The ALE analysis results of 32 studies on voice identity processing (*i.e.,* analysis ) are shown in Fig. 2A and Table 3. Significant activation was noted in two large clusters distributed on both sides of the brain in the temporal lobe, including the bilateral STG, the bilateral MTG, the left inferior temporal gyrus (ITG), and the right fusiform gyrus (FG), and in

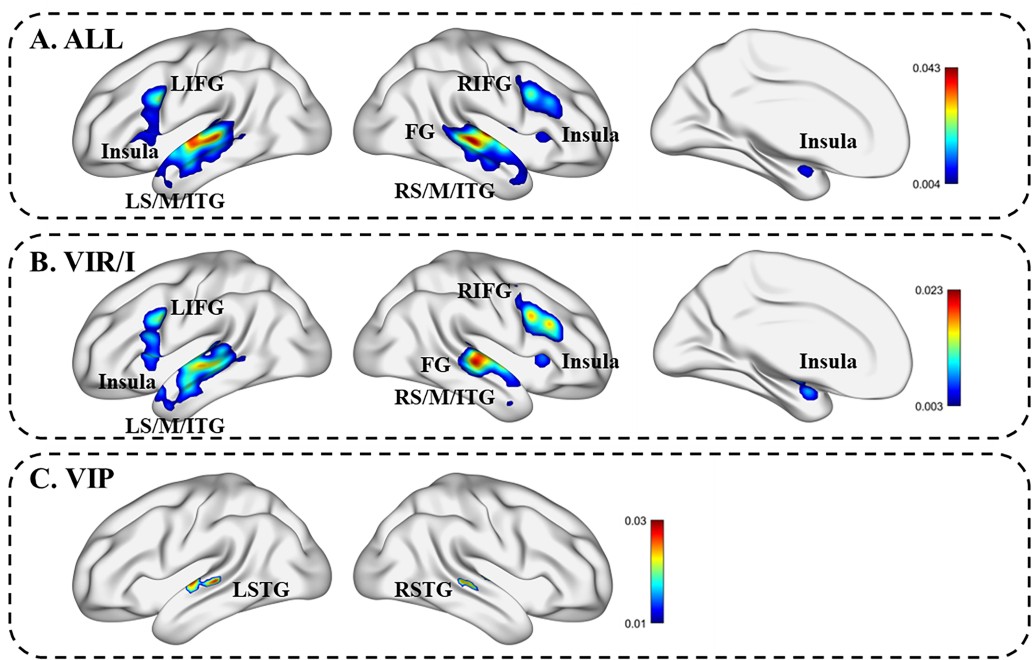

**Figure 2 Summary of ALE activation results.** (A) ALE activation map of voice identity processing. (B) ALE activation map of voice identity recognition/identification. (C) ALE activation map of voice identity perception. LIFG, left inferior frontal gyrus; RIFG, right inferior frontal gyrus; LSTG, left superior temporal gyrus; LMTG, left middle temporal gyrus; LITG, left inferior temporal gyrus; RSTG, right superior temporal gyrus; RMTG, right middle temporal gyrus; RITG, right inferior temporal gyrus; FG, fusiform gyrus.

the frontal lobe, including the bilateral IFG, the right middle frontal gyrus (MFG), the left parahippocampal gyrus, the bilateral insula, the right claustrum and the right caudate nucleus.

## Voice identity recognition/identification

For the ALE analysis of 22 VIR and VII (*i.e.,* analysis ), significant activation was found in 3 clusters, including the temporal lobe (*e.g.,* the bilateral STG, the bilateral MTG, the left ITG, and the right FG), the frontal lobe (*e.g.,* the bilateral IFG and the right MFG), the left parahippocampal gyrus, the bilateral insula, and so on (for details, see Table 4 and Fig. 2B).

## Voice identity perception

As shown in Fig. 2C and Table 5, the results of the ALE analysis of VIP revealed significant activation in the bilateral STG.

## Comparison and conjoint analysis

The ALE analysis of VIR/I *versus* VIP, that is, (R + I) −P, showed significant activation in the right IFG and MFG (see Fig. 3A and Table 6). Conversely, the ALE analysis of VIP *versus* VIR/I, *i.e.,* P−(R + I), showed significant activation in the bilateral STG, the bilateral MTG and the left IFG (see Fig. 3B and Table 6).

**Table 3  Brain regions of voice identity processing.**

| H | Cluster size (mm³) | Region | BA | X | Y | Z | ALE (×10⁻²) |
|---|---|---|---|---|---|---|---|
| R | 15285 | Superior Temporal Gyrus | 22 | 54 | −16 | 2 | 4.33 |
|  |  |  |  | 56 | −24 | 0 | 4.09 |
|  |  |  |  | 54 | −2 | 0 | 3.48 |
|  |  |  | 41 | 46 | −40 | 6 | 1.59 |
|  |  |  | 38 | 44 | 8 | −10 | 1.08 |
|  | 438 | Fusiform Gyrus | 37 | 36 | −38 | −12 | 0.67 |
|  | 8227 | Middle Temporal Gyrus | 21 | 54 | 8 | −22 | 1.20 |
|  |  |  |  | 56 | 4 | −26 | 1.02 |
|  | 3291 | Middle Frontal Gyrus | 9 | 46 | 14 | 28 | 2.65 |
|  |  |  | 46 | 46 | 24 | 24 | 2.06 |
|  |  |  | 6 | 38 | 2 | 40 | 1.04 |
|  | 1682 | Inferior Frontal Gyrus | 45 | 40 | 26 | 4 | 0.98 |
|  | * | Claustrum | * | 28 | 20 | 6 | 1.04 |
|  | 2998 | Insula | 13 | 42 | 12 | -2 | 0.80 |
|  |  |  | 13 | 46 | 10 | 0 | 0.76 |
|  | * | Caudate | * | 36 | −34 | −4 | 0.71 |
| L | 10362 | Superior Temporal Gyrus | 22 | −56 | −10 | 0 | 3.71 |
|  |  |  |  | −58 | −26 | 2 | 3.68 |
|  |  |  |  | −56 | −14 | 4 | 3.64 |
|  |  |  | 21 | −60 | −18 | −2 | 3.55 |
|  |  |  | 38 | −52 | 0 | −6 | 1.95 |
|  |  |  |  | −50 | 10 | −18 | 0.88 |
|  |  |  |  | −48 | 14 | −20 | 0.86 |
|  |  |  | 41 | −42 | −34 | 6 | 1.25 |
|  | 8954 | Middle Temporal Gyrus | 21 | −54 | 6 | −28 | 0.99 |
|  | 1308 | Inferior Temporal Gyrus | 20 | −62 | −18 | −20 | 0.64 |
|  | 1844 | Transverse Temporal Gyrus | 41 | −44 | −22 | 12 | 0.95 |
|  | 4124 | Inferior Frontal Gyrus | 9 | −44 | 10 | 26 | 2.62 |
|  |  |  | 45 | −38 | 26 | 4 | 1.57 |
|  | 2750 | Insula | 13 | −46 | 12 | 4 | 1.44 |
|  |  |  |  | −30 | 18 | 8 | 1.18 |
|  | 738 | Parahippocampal Gyrus | * | −28 | −2 | −16 | 1.06 |

**Notes.**

H, hemisphere; BA, Brodmann area; ALE, GingerALE operation value.

Moreover, through the conjoint ALE analysis of VIR/I and VIP, *i.e.,* (R + I) + P, the results showed significantly distributed activation in the bilateral STG (see Fig. 3C and Table 6).

## DISCUSSION

In the present study, we reorganized and defined the different levels of voice processing and adopted the meta-analysis method to distinguish the neural mechanism of voice identity processing with different levels of familiarity. Based on the ALE meta-analysis of 32 studies,

**Table 4  Brain regions for voice identity recognition/identification.**

| H | Cluster size (mm³) | Region | BA | X | Y | Z | ALE ($\times 10^{-2}$) |
|---|---|---|---|---|---|---|---|
| R | 10919 | Superior Temporal Gyrus | * | 58 | −22 | 2 | 2.06 |
| | | | 22 | 46 | −20 | −2 | 1.97 |
| | | | | 54 | −4 | 0 | 1.73 |
| | | | 38 | 44 | 18 | −18 | 0.86 |
| | | | | 54 | 12 | −14 | 0.76 |
| | | | | 56 | 10 | −22 | 0.72 |
| | | | | 52 | 14 | −30 | 0.61 |
| | 278 | Fusiform Gyrus | 37 | 36 | −38 | −12 | 0.67 |
| | 4658 | Middle Temporal Gyrus | 21 | 62 | −6 | −10 | 1.08 |
| | | | | 58 | 2 | −28 | 0.95 |
| | 1389 | Transverse Temporal Gyrus | 41 | 44 | −28 | 10 | 0.71 |
| | 5251 | Middle Frontal Gyrus | 9 | 48 | 16 | 28 | 2.30 |
| | | | | 38 | 12 | 30 | 1.87 |
| | | | 46 | 46 | 24 | 24 | 2.06 |
| | | | 6 | 38 | 2 | 40 | 1.04 |
| | 2543 | Inferior Frontal Gyrus | 44 | 60 | 16 | 16 | 0.67 |
| | 214 | Claustrum | * | 28 | 20 | 6 | 1.04 |
| | 2735 | Insula | 13 | 42 | 12 | 0 | 0.78 |
| | | | 13 | 46 | 10 | 0 | 0.76 |
| | * | Caudate | * | 36 | −34 | −4 | 0.71 |
| L | 8964 | Superior Temporal Gyrus | 22 | −54 | −14 | 6 | 1.61 |
| | | | | −48 | −32 | 6 | 1.14 |
| | | | | −46 | −2 | −6 | 0.85 |
| | | | 41 | −42 | −34 | 6 | 1.13 |
| | | | 42 | −62 | −30 | 8 | 1.04 |
| | | | 38 | −50 | 10 | −18 | 0.87 |
| | | | | −48 | 14 | −20 | 0.86 |
| | 6811 | Middle Temporal Gyrus | 21 | −60 | −16 | −4 | 2.02 |
| | | | | −54 | 6 | −28 | 0.99 |
| | | | 22 | −60 | −40 | 6 | 0.68 |
| | 1414 | Inferior Temporal Gyrus | 21 | −56 | −8 | −12 | 1.33 |
| | | | 20 | −62 | −18 | −20 | 0.63 |
| | 1414 | Transverse Temporal Gyrus | 41 | −44 | −22 | 12 | 0.95 |
| | 3759 | Inferior Frontal Gyrus | 9 | −42 | 6 | 28 | 1.44 |
| | | | | −50 | 18 | 20 | 1.01 |
| | 3084 | Insula | 13 | −46 | 12 | 4 | 1.44 |
| | | | | −30 | 18 | 8 | 1.16 |
| | 1896 | Parahippocampal Gyrus | * | −28 | −2 | −16 | 1.06 |
| | | | | −20 | −8 | −12 | 0.97 |

**Notes.**

H, hemisphere; BA, Brodmann area; ALE, GingerALE operation value.

**Table 5  Brain regions for voice identity perception.**

| L/R | Cluster size (mm³) | Region | BA | X | Y | Z | ALE (×10⁻²) |
|-----|--------------------|--------|-----|-----|-----|-----|-------------|
| R | 542 | Superior Temporal Gyrus | 22 | 54 | −16 | 2 | 2.87 |
| | | | | 56 | −26 | 2 | 2.46 |
| | | | | 54 | −30 | 4 | 2.43 |
| L | 820 | Superior Temporal Gyrus | 22 | −58 | −26 | 4 | 2.64 |
| | | | | −56 | −10 | 2 | 2.56 |

**Notes.**

H, hemisphere; BA, Brodmann area; ALE, GingerALE operation value.

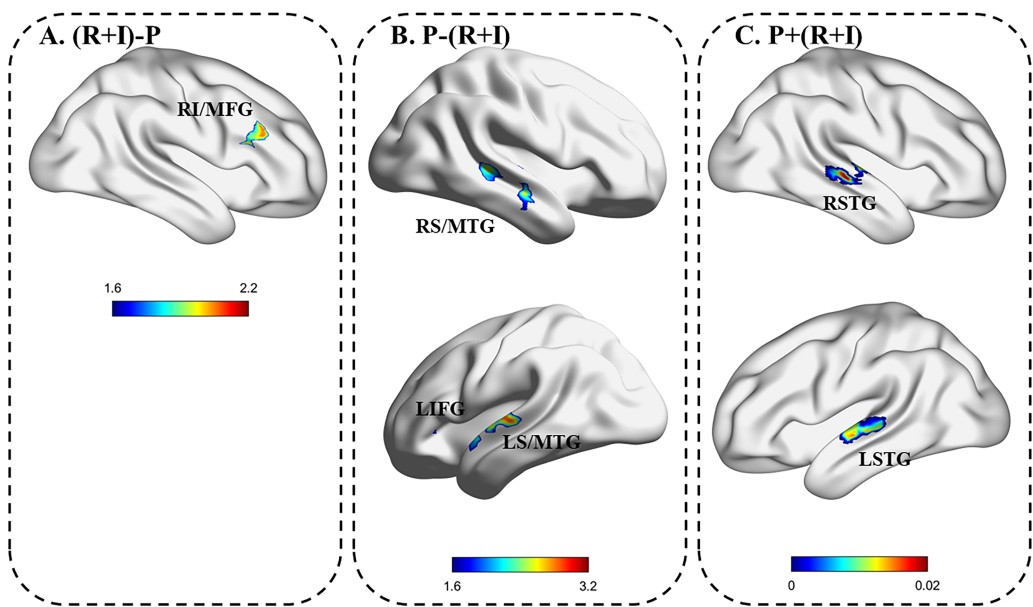

**Figure 3  Comparison and conjoint analysis results.** (A) Voice identity recognition/identification *vs.* voice identity perception. (B) Voice identity perception *vs.* voice identity recognition/recognition. (C) Voice identity recognition/recognition combined with voice identity perception. LIFG, left inferior frontal gyrus; RIFG, right inferior frontal gyrus; LSTG, left superior temporal gyrus; LMTG, left middle temporal gyrus; RSTG, right superior temporal gyrus; RMTG, right middle temporal gyrus.

we found that the core area of voice identity processing is located in the bilateral superior temporal gyrus, which is involved in voice identity processing regardless of whether the voice is familiar. Some brain regions of the frontal-temporal lobe were involved in varying degrees of voice identity processing with different familiarities.

## Brain mechanism of unfamiliar voice processing

As mentioned above, voice perception studies have repeatedly demonstrated the existence of voice-selective sensitive regions in the human temporal auditory cortex. Recently, *Pernet et al. (2015)* proposed a voice localizer scan to pinpoint the voice-sensitive areas of the temporal lobe, termed temporal voice areas (TVAs), which were mainly located in the bilateral STG/S. Of note, the TVA region is commonly associated with species-specific

**Table 6  Comparison and conjoint analysis.**

| Type | L/R | Cluster size (mm³) | Region | BA | X | Y | Z |
|---|---|---|---|---|---|---|---|
| (R+I)-P | R | 1053 | Middle Frontal Gyrus | 46 | 40.8 | 26.9 | 24.6 |
| | | | | | 48.5 | 29.5 | 24 |
| | | | | | 42 | 21.3 | 21 |
| | | | | 9 | 46.5 | 28.5 | 29 |
| | | 1459 | Inferior Frontal Gyrus | 45 | 49.2 | 24 | 21.1 |
| | | | | 44 | 56 | 18 | 16 |
| | | | | | 60 | 18.7 | 15.3 |
| | | | | 9 | 52 | 20 | 20 |
| | | | | | 55.3 | 16.9 | 24 |
| P-(R+I) | R | 1187 | Superior Temporal Gyrus | * | 54 | −18 | 2 |
| | | 1936 | Middle Temporal Gyrus | 22 | 52 | −34 | 6 |
| | | | | 21 | 52 | −14 | −10 |
| | | | | | 64 | −18 | −10 |
| | L | 1498 | Superior Temporal Gyrus | * | −64 | −26 | 2 |
| | | | | 22 | −62 | −19 | 6 |
| | | | | | −58 | −12 | 0 |
| | | | | | −62 | −12 | 4 |
| | | 418 | Middle Temporal Gyrus | 21 | −58 | −2 | −6 |
| | | 264 | Inferior Frontal Gyrus | 47 | −38.4 | 28.7 | 0.7 |
| (R+I)+P | R | 554 | Superior Temporal Gyrus | 21 | 56 | −22 | 0 |
| | | | | 22 | 58 | −16 | 4 |
| | L | 820 | Superior Temporal Gyrus | 21 | −60 | −20 | −2 |
| | | | | | −60 | −12 | −2 |
| | | | | 22 | −54 | −14 | 4 |

**Notes.**

H, hemisphere; BA, Brodmann area.

sensitivity of the human voice (*Fecteau et al., 2004*). In the present study, ALE analysis of VIP showed activation of the bilateral STG, which overlaps with TVA regions, suggesting that the bilateral STG is sensitive to human voice and responsible for processing acoustic information in voices. Combined with the concept of VIP we defined, that is, the listener only knows that the voice stimulus is the sound produced by the vibration of human vocal cords but does not know "who" it is. Thus, the voice identity is unfamiliar to the listener. In other words, the brain regions activated by the perception of a stranger's voice can complete the basic acoustic processing of the unfamiliar voice input stimulus but cannot evoke the processing of the specific "identity" corresponding to the stimulus.

Of course, this could also be a result of the task or paradigm, since most voice perception experiments involve subjects passively listening to one stimulus after another (*Fecteau et al., 2004*; *Fecteau et al., 2005*; *Warren et al., 2006*); therefore, subjects do not have the opportunity to respond to the current stimulus and thus could not further compare the current input stimulus with various "voice identities" stored in their memory. In contrast, *Roswandowitz, Swanborough & Fruehholz (2021)* added the task of actively judging whether

the current stimulus was a human voice and found that the IFG still had functional involvement when the voice was actively processed. In addition, activation of the IFG was found in two recent studies of VIP with large samples (*Aglieri et al., 2018*; *Pernet et al., 2015*), different from the active task of *Roswandowitz, Swanborough & Fruehholz (2021)*. These two large-sample studies both reported activation of the inferior frontal gyrus under the condition of passive listening. They concluded that due to considerable interindividual variability in frontal lobe anatomy and function, frontal lobe activation was understandable in some subjects under large sample conditions. At the same time, voice is closely related to speech and mixed with certain language content, so part of the frontal lobe can be used as the voice-sensitive extension area outside the temporal lobe.

Therefore, the current ALE analysis results of VIP should be interpreted with caution given the passive listening design adopted in most experiments, considering the difference in subject sample size, or based on whether the stimulus materials contain language information.

## Brain mechanism of voice processing in familiar people

The activation of the superior temporal cortex (STC) in the transverse temporal gyrus extending to the superior temporal gyrus was consistent with previous studies, and this region was also consistent with the classic TVA region (*Belin et al., 2000*; *Pernet et al., 2015*), suggesting that this region activates and participates in acoustic processing of a voice regardless of voice familiarity (*Andics et al., 2010*; *Belin et al., 2011*). The activation of the MTG is consistent with the findings of *Bethmann & Brechmann (2014)* and *Bethmann, Scheich & Brechmann (2012)*, who assessed the degree of voice-specific processing in the superior/middle temporal cortex and found that the difference between the specificity of voice processing and other sounds increased from the superior temporal gyrus to the middle temporal gyrus. In the current study, activation of the MTG may expand the TVA region in the classical temporal lobe, suggesting that the vocal selective brain region includes the area from the STC to the MTG, which is responsible for the basic acoustic processing of input stimuli.

Moreover, the ITG has been recruited to process the identity of familiar speakers. A previous study found that a patient with bilateral temporal lobe atrophy and more obvious atrophy of the right ITG (including the fusiform gyrus) had severe defects in familiar voice identification (*Roswandowitz, Maguinness & Von Kriegstein, 2018*). As the homologous contralateral brain region of the right ITG, the role of the left ITG in the identification process of familiar voice needs further investigation. In addition, as another possible explanation for the activation of the left ITG, it has been found that the ITG is activated during tone tasks (*Stevens, 2004*), and a voice contains a large amount of spectral information (*Lattner, Meyer & Friederici, 2005*). The left ITG may participate in the spectral information processing of a voice together with the STC.

Consistent with previous findings on high-familiarity voice identity processing (*Hoelig et al., 2017*; *Joassin, Maurage & Campanella, 2011*; *Shah et al., 2001*; *Von Kriegstein et al., 2005*), the activation in the right FG suggested that it may be involved in ascertaining a highly familiar voice identity, especially when the information of the voice identity has multimodal

semantic associations. As *Von Kriegstein et al. (2005)* found, when subjects recognized the voice identity of a completely familiar person, there was functional interaction between the face recognition area and the voice-sensitive area, indicating that listeners might automatically retrieve the personal identity corresponding to the voice. In addition, it was found that in the process of familiar voice identification in early blind patients, there was functional connectivity between the voice selection area of the left temporal lobe and the right FG according to *Dormal et al. (2017)*. Moreover, the activation of the FG was affected by the consistency of voice identity and showed more activity during the recognition of familiar voices for advanced or congenital blindness (*Hoelig et al., 2014a*; *Hoelig et al., 2014b*). Evidence from both sighted and blind people suggests an integral role for the fusiform gyrus in the process of identity recognition/identification of familiar voices.

Importantly, extensive frontal lobe regions (*e.g.*, the bilateral IFG and the right MFG) were involved in high-familiarity voice identity processing, which is consistent with these previous studies (*Aglieri et al., 2021*; *Andics, McQueen & Petersson, 2013*; *Blank, Wieland & Von Kriegstein, 2014*; *Latinus, Crabbe & Belin, 2011*; *Rama et al., 2004*; *Relander & Rama, 2009*; *Zaeske, Hasan & Belin, 2017*), when the voice stimuli were trained such that they reached a degree of familiarity. In addition, studies of voice prototypes have found that bilateral IFG activation is often associated with acoustic information processing of a voice (*Andics et al., 2010*). Furthermore, the activation of the frontal lobe may be a function of cognitive control in voice identity, and the IFG and MFG are highly correlated with cognitive control functions. This finding may indicate that subjects need to inhibit the interference information, then employ task-related resources to retrieve personal information in the memory system and compare it with the voice currently being heard, and finally complete voice identity recognition and identification (*Hu et al., 2017*).

The hippocampus is commonly associated with memory function (*Schacter et al., 1996*; *Stark & Squire, 2001*) and is also associated with participants' familiarity with voice identity (*Shah et al., 2001*) or retention of working memory for voice identity (*Rama & Courtney, 2005*). Therefore, the hippocampus may be related to the memory content corresponding to the retrieval of a familiar voice identity. In addition, it has been found that the bilateral insula and the bilateral subfrontal cortex are activated in the processing of acoustic information of a voice (*Andics et al., 2010*; *Latinus, Crabbe & Belin, 2011*). Pertinently, the insula is also associated with working memory and plays an important role in maintaining and recognizing the identity of familiar speakers' voices (*Rama et al., 2004*; *Relander & Rama, 2009*). It is worth noting that the activation of the claustrum and caudate nucleus has never been reported in previous studies. Structurally, the claustrum is adjacent to the lentiform nucleus, but its function is unknown. The caudate nucleus is closely connected to the lentiform nucleus, and both are part of the striatum, which in vertebrates is mainly involved in the execution of fine movements (*Grillner et al., 2005*). In this study, we found that the activation of the claustrum and caudate nucleus in voice processing, one possibility may be the result of the concomitant activation of the insula, which may not be associated with voice processing itself.

Together, the ALE analysis results showed that VIR/I involves a wide range of brain networks, including basic acoustic analysis, multimodal information retrieval, cognitive regulation, memory retrieval and other functions.

### Similarities and differences in voice identity processing mechanisms between familiar speakers and unfamiliar speakers
*Differences in the brain mechanism of voice identity processing between familiar and unfamiliar speakers*

The comparative ALE analysis showed that there were significant differences in brain activation patterns between familiar speakers and unfamiliar speakers in voice identity processing.

First, the results of (R + I)−P showed that familiar voice recognition/identification significantly activated the right MFG and IFG compared with the activation noted during unfamiliar voice perception. Previous studies have found that the right MFG and IFG were significantly activated during familiar voice recognition/identification tasks (*Latinus, Crabbe & Belin, 2011*; *Nakamura et al., 2001*; *Shah et al., 2001*; *Zaeske, Hasan & Belin, 2017*). In addition, the MFG and IFG were recruited in the maintenance of acoustic memory of voice information based on the delay-matching paradigm of voice identity recognition (*Rama et al., 2004*). Combined with previous studies, in our meta-analysis, the right MFG and IFG may be particularly involved in familiar voice identity processing compared with unfamiliar voice identity processing (*Von Kriegstein et al., 2005*). Notably, as described in Section 4.2 of this paper, the involvement of the right frontal lobe may represent the involvement of cognitive function (*Hu et al., 2017*), which may assist in extracting identity information about familiar speakers based on the acoustic analysis outcome. However, this meta-analysis did not reveal activation of the left IFG, which has been activated in previously familiar voice studies (*Aglieri et al., 2021*; *Shah et al., 2001*).

Second, P−(R +I) showed significant activation in the bilateral STG, bilateral MTG, and left IFG. One possible explanation for bilateral temporal lobe activation is that listeners conduct more acoustic spectrum analysis of unfamiliar voices that have never been heard before, and these unfamiliar voices cannot provide any useful identity information in memory. Thus, the listener only conducts voice-specific processing and judges whether the voices are human based on the current stimulus (*Belin et al., 2000*); meanwhile, the temporal lobe is activated as a specific and typical voice-sensitive region in the human brain.

Notably, although the results of (R +I)−P did not show more intense activation in the left IFG, P−(R +I) showed several activated regions in the left IFG located in BA47. It is generally believed that left BA 47 is more engaged in semantic processing (*Hagoort, 2005*; *Hagoort, 2013*), and a meta-analysis of the IFG also found that the left IFG is enrolled in processing semantic information (*Belyk et al., 2017*). Considering that the above (R +I)−P analysis showed activation only in the right MFG and IFG but no activation in the left frontal lobe, one possible explanation is that the function of cognitive control could be recruited for listeners to perform VIR/I. Particularly, the results of P−(R +I) showed more left frontal lobe activation, which may indicate that listeners are unable to process

the identity corresponding to the unfamiliar voice during perception, so they implicitly process the linguistic information in the stimulus that then leads to the involvement of the left frontal lobe.

### Shared brain mechanism in voice identity processing between familiar and unfamiliar speakers

Conjoint ALE analysis showed that the bilateral STG was the common neural basis for the three levels of voice identity processing. As a classic TVA region, the bilateral STG/S is always more responsive to a human voice than to various sound stimuli, such as environmental noise and animal noises (*Agus et al., 2017*; *Belin et al., 2011*; *Belin et al., 2000*; *Formisano et al., 2008*; *Pernet et al., 2015*). In the present study, the bilateral STG showed significant activation in the conjoint analysis of familiar and unfamiliar voices, which indicates that the bilateral STG is an essential brain region for the acoustic analysis of voice identity processing regardless of the degree of familiarity of the voice (*Roswandowitz et al., 2018*).

### Limitations

As mentioned in the introduction, the recent research on voice identity processing has mixed terminology. At the same time, the field of VIP intersects with the field of voice emotion perception, so the data related to identify processing may be affected by other factors, which then leads to considerable complications in literature retrieval and coordinate data sorting.

Although this study encompassed the relevant literature as much as possible, after identifying the relevant research (*Eickhoff et al., 2016*), it was found that the number of studies corresponding to the three subcategories, which included VIP, VIR and VII, was still insufficient to meet the requirements of ALE analysis based on cluster-FWE correction alone. Therefore, it was impossible to compare the three categories separately; otherwise, results with very low statistical power would be obtained, which is a weakness of this study. We are not sure whether there is statistical bias in this method, but we have improved the statistical efficiency in ALE analysis as much as possible. It is expected that a large number of well-defined studies and data will be used to supplement and improve upon this study in the future; in addition, we will continue to assess relevant studies. At the same time, we will further expand the search scope and sort out the relevant literature in these three categories, with the goal of further improving on the present study. In the future, we hope to conduct a series of experiments to compare the brain activation of voice identities with different degrees of familiarity (perception, recognition and identification) to further verify the results of this meta-analysis.

## CONCLUSION

In summary, this study is the first to define three types of voice identity processing: VIP, VIR and VII. An ALE analysis was performed on 32 studies of "VIP" and "VIR/I" to reveal the similarities and differences in neural representation between unfamiliar voice identity and familiar voice identity processing. These findings suggest that the bilateral STG is the core area of voice identity processing and is not affected by the listener's familiarity with

the speaker's identity. The bilateral STG, MTG and left IFG were more active in the process of unfamiliar voice identity perception than in familiar voice identity perception. The right inferior/middle frontal gyrus, right fusiform gyrus, left parahippocampal gyrus and bilateral insula were more active in the process of familiar voice recognition/identification than in unfamiliar voice identity identification.

## ACKNOWLEDGEMENTS

The authors thank all the researchers contacted for their assistance with this research, especially Professors Pascal Belin, Simon B. Eickhoff and Mick Fox for their support and helpful suggestions.

### Funding

This work was supported by grants from the National Natural Science Foundation of China (31900750) to Xueping Hu and the Natural Science Research Foundation of Jiangsu Normal University (18XLRX011) to Xueping Hu. The funders had no role in study design, data collection and analysis, decision to publish, or preparation of the manuscript.

### Grant Disclosures

The following grant information was disclosed by the authors:
National Natural Science Foundation of China: 31900750.
Natural Science Research Foundation of Jiangsu Normal University: 18XLRX011.

### Competing Interests

The authors declare there are no competing interests.

### Author Contributions

- YuXiang Sun conceived and designed the experiments, performed the experiments, analyzed the data, prepared figures and/or tables, authored or reviewed drafts of the article, and approved the final draft.
- Lili Ming performed the experiments, authored or reviewed drafts of the article, and approved the final draft.
- Jiamin Sun performed the experiments, authored or reviewed drafts of the article, and approved the final draft.
- FeiFei Guo performed the experiments, authored or reviewed drafts of the article, and approved the final draft.
- Qiufeng Li performed the experiments, authored or reviewed drafts of the article, and approved the final draft.
- Xueping Hu conceived and designed the experiments, performed the experiments, analyzed the data, prepared figures and/or tables, authored or reviewed drafts of the article, and approved the final draft.

## Data Availability

All original coordinate data are available in the Supplemental Files.

## Supplemental Information

Supplemental information for this article can be found online at http://dx.doi.org/10.7717/peerj.14976#supplemental-information.

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

# PeerJ

**Birkett PB, Hunter MD, Parks RW, Farrow TF, Lowe H, Wilkinson LD, Woodruff PW. 2007.** Voice familiarity engages auditory cortex. *Neuroreport* **18(13)**:1375–1378 DOI 10.1097/WNR.0b013e3282aa43a3.

**Blank H, Wieland N, Von Kriegstein K. 2014.** Person recognition and the brain: merging evidence from patients and healthy individuals. *Neuroscience and Biobehavioral Reviews* **47**:717–734 DOI 10.1016/j.neubiorev.2014.10.022.

**Bonte M, Hausfeld L, Scharke W, Valente G, Formisano E. 2014.** Task-dependent decoding of speaker and vowel identity from auditory cortical response patterns. *Journal of Neuroscience* **34(13)**:4548–4557 DOI 10.1523/jneurosci.4339-13.2014.

**Dormal G, Pelland M, Rezk M, Yakobov E, Lepore F, Collignon O. 2017.** Functional Preference for Object Sounds and Voices in the Brain of Early Blind and Sighted Individuals. *Journal of Cognitive Neuroscience* **30(1)**:86–106 DOI 10.1162/jocn_a_01186.

**Eickhoff SB, Bzdok D, Laird AR, Kurth F, Fox PT. 2012.** Activation likelihood estimation meta-analysis revisited. *Neuroimage* **59(3)**:2349–2361 DOI 10.1016/j.neuroimage.2011.09.017.

**Eickhoff SB, Laird AR, Fox PM, Lancaster JL, Fox PT. 2017.** Implementation errors in the GingerALE software: Description and recommendations. *Human Brain Mapping* **38(1)**:7–11 DOI 10.1002/hbm.23342.

**Eickhoff SB, Laird AR, Grefkes C, Wang LE, Zilles K, Fox PT. 2009.** Coordinate-based activation likelihood estimation meta-analysis of neuroimaging data: a random-effects approach based on empirical estimates of spatial uncertainty. *Human Brain Mapping* **30(9)**:2907–2926 DOI 10.1002/hbm.20718.

**Eickhoff SB, Nichols TE, Laird AR, Hoffstaedter F, Amunts K, Fox PT, Bzdok D, Eickhoff CR. 2016.** Behavior, sensitivity, and power of activation likelihood estimation characterized by massive empirical simulation. *Neuroimage* **137**:70–85 DOI 10.1016/j.neuroimage.2016.04.072.

**Fecteau S, Armony JL, Joanette Y, Belin P. 2004.** Is voice processing species-specific in human auditory cortex?—An fMRI study. *Neuroimage* **23(3)**:840–848 DOI 10.1016/j.neuroimage.2004.09.019.

**Fecteau S, Armony JL, Joanette Y, Belin P. 2005.** Sensitivity to voice in human prefrontal cortex. *Journal of Neurophysiology* **94(3)**:2251–2254 DOI 10.1152/jn.00329.2005.

**Formisano E, De Martino F, Bonte M, Goebel R. 2008.** Who is saying What? Brain-based decoding of human voice and speech. *Science* **322(5903)**:970–973 DOI 10.1126/science.1164318.

**Grillner S, Helligren J, Menard A, Saitoh K, Wikstrom MA. 2005.** Mechanisms for selection of basic motor programs - roles for the striatum and pallidum. *Trends in Neurosciences* **28(7)**:364–370 DOI 10.1016/j.tins.2005.05.004.

**Hagoort P. 2005.** On Broca, brain, and binding: a new framework. *Trends in Cognitive Sciences* **9(9)**:416–423 DOI 10.1016/j.tics.2005.07.004.
**Hagoort P. 2013.** MUC (Memory, Unification, Control) and beyond. *Frontiers in Psychology* **4**:416–416 DOI 10.3389/fpsyg.2013.00416.

**Hasan BAS, Valdes-Sosa M, Gross J, Belin P. 2016.** Hearing faces and seeing voices: amodal coding of person identity in the human brain. *Scientific Reports* **6**:37494 DOI 10.1038/srep37494.

**Hoelig C, Foecker J, Best A, Roeder B, Buechel C. 2014a.** Brain systems mediating voice identity processing in blind humans. *Human Brain Mapping* **35**(**9**):4607–4619 DOI 10.1002/hbm.22498.

**Hoelig C, Foecker J, Best A, Roeder B, Buechel C. 2014b.** Crossmodal plasticity in the fusiform gyrus of late blind individuals during voice recognition. *Neuroimage* **103**:374–382 DOI 10.1016/j.neuroimage.2014.09.050.

**Hoelig C, Foecker J, Best A, Roeder B, Buechel C. 2017.** Activation in the angular Gyrus and in the pSTS is modulated by face primes during voice recognition. *Human Brain Mapping* **38**(**5**):2553–2565 DOI 10.1002/hbm.23540.

**Hu X, Wang X, Gu Y, Luo P, Yin S, Wang L, Fu C, Qiao L, Du Y, Chen A. 2017.** Phonological experience modulates voice discrimination: evidence from functional brain networks analysis. *Brain and Language* **173**:67–75 DOI 10.1016/j.bandl.2017.06.001.

**Joassin F, Maurage P, Campanella S. 2011.** The neural network sustaining the cross-modal processing of human gender from faces and voices: an fMRI study. *Neuroimage* **54**(**2**):1654–1661 DOI 10.1016/j.neuroimage.2010.08.073.

**Kuhl PK. 2011.** Who's talking? *Science* **333**(**6042**):529–530 DOI 10.1126/science.1210277.

**Latinus M, Crabbe F, Belin P. 2011.** Learning-induced changes in the cerebral processing of voice identity. *Cerebral Cortex* **21**(**12**):2820–2828 DOI 10.1093/cercor/bhr077.

**Latinus M, McAleer P, Bestelmeyer PEG, Belin P. 2013.** Norm-based coding of voice identity in human auditory cortex. *Current Biology* **23**(**12**):1075–1080 DOI 10.1016/j.cub.2013.04.055.

**Lattner S, Meyer ME, Friederici AD. 2005.** Voice perception: sex, pitch, and the right hemisphere. *Human Brain Mapping* **24**(**1**):11–20 DOI 10.1002/hbm.20065.

**Lee YS, Peelle JE, Kraemer D, Lloyd S, Granger R. 2015.** Multivariate sensitivity to voice during auditory categorization. *Journal of Neurophysiology* **114**(**3**):1819–1826 DOI 10.1152/jn.00407.2014.

**Maguinness C, Roswandowitz C, Von Kriegstein K. 2018.** Understanding the mechanisms of familiar voice-identity recognition in the human brain. *Neuropsychologia* **116**:179–193 DOI 10.1016/j.neuropsychologia.2018.03.039.

**Mathiak K, Menning H, Hertrich I, Mathiak KA, Zvyagintsev M, Ackermann H. 2007.** Who is telling what from where? A functional magnetic resonance imaging study. *Neuroreport* **18**(**5**):405–409 DOI 10.1097/WNR.0b013e328013cec4.

**Mathias SR, Von Kriegstein K. 2014.** How do we recognise who is speaking? *Frontiers in bioscience* **6**(**1**):92–109.

Nakamura K, Kawashima R, Sugiura M, Kato T, Nakamura A, Hatano K, Nagumo S, Kubota K, Fukuda H, Ito K, Kojima S. 2001. Neural substrates for recognition of familiar voices: a PET study. *Neuropsychologia* **39(10)**:1047–1054 DOI 10.1016/s0028-3932(01)00037-9.

Ogg M, Moraczewski D, Kuchinsky SE, Slevc LR. 2019. Separable neural representations of sound sources: speaker identity and musical timbre. *Neuroimage* **191**:116–126 DOI 10.1016/j.neuroimage.2019.01.075.

Parola A, Simonsen A, Bliksted V, Fusaroli R. 2020. Voice patterns in schizophrenia: a systematic review and Bayesian meta-analysis. *Schizophrenia Research* **216**:24–40 DOI 10.1016/j.schres.2019.11.031.

Pernet CR, McAleer P, Latinus M, Gorgolewski KJ, Charest I, Bestelmeyer PEG, Watson RH, Fleming D, Crabbe F, Valdes-Sosa M, Belin P. 2015. The human voice areas: spatial organization and inter-individual variability in temporal and extra-temporal cortices. *Neuroimage* **119**:164–174 DOI 10.1016/j.neuroimage.2015.06.050.

Rama P, Courtney SM. 2005. Functional topography of working memory for face or voice identity. *Neuroimage* **24(1)**:224–234 DOI 10.1016/j.neuroimage.2004.08.024.

Rama P, Poremba A, Sala JB, Yee L, Malloy M, Mishkin M, Courtney SM. 2004. Dissociable functional cortical topographies for working memory maintenance of voice identity and location. *Cerebral Cortex* **14(7)**:768–780 DOI 10.1093/cercor/bhh037.

Relander K, Rama P. 2009. Separate neural processes for retrieval of voice identity and word content in working memory. *Brain Research* **1252**:143–151 DOI 10.1016/j.brainres.2008.11.050.

Roswandowitz C, Kappes C, Obrig H, Von Kriegstein K. 2018. Obligatory and facultative brain regions for voice-identity recognition. *Brain* **141**:234–247 DOI 10.1093/brain/awx313.

Roswandowitz C, Maguinness C, Von Kriegstein K. 2018. Deficits in voice-identity processing: acquired and developmental phonagnosia. *Preprints 2018* DOI 10.20944/preprints201806.0280.v1.

Roswandowitz C, Swanborough H, Fruehholz S. 2021. Categorizing human vocal signals depends on an integrated auditory-frontal cortical network. *Human Brain Mapping* **42(5)**:1503–1517 DOI 10.1002/hbm.25309.

Schacter DL, Alpert NM, Savage CR, Rauch SL, Albert MS. 1996. Conscious recollection and the human hippocampal formation: evidence from positron emission tomography. *Proceedings of the National Academy of Sciences of the United States of America* **93(1)**:321–325 DOI 10.1073/pnas.93.1.321.

Scott SK. 2019. From speech and talkers to the social world: the neural processing of human spoken language. *Science* **366(6461)**:58–61 DOI 10.1126/science.aax0288.

**Shah NJ, Marshall JC, Zafiris O, Schwab A, Zilles K, Markowitsch HJ, Fink GR. 2001.** The neural correlates of person familiarity—a functional magnetic resonance imaging study with clinical implications. *Brain* **124**:804–815 DOI 10.1093/brain/124.4.804.

**Stark CEL, Squire LR. 2001.** Simple and associative recognition memory in the hippocampal region. *Learning & Memory* **8(4)**:190–197 DOI 10.1101/lm.40701.

**Stevens AA. 2004.** Dissociating the cortical basis of memory for voices, words and tones. *Cognitive Brain Research* **18(2)**:162–171 DOI 10.1016/j.cogbrainres.2003.10.008.

**Turkeltaub PE, Eden GF, Jones KM, Zeffiro TA. 2002.** Meta-analysis of the functional neuroanatomy of single-word reading: method and validation. *Neuroimage* **16(3)**:765–780 DOI 10.1006/nimg.2002.1131.

**Turkeltaub PE, Eickhoff SB, Laird AR, Fox M, Wiener M, Fox P. 2012.** Minimizing within-experiment and within-group effects in activation likelihood estimation meta-analyses. *Human Brain Mapping* **33(1)**:1–13 DOI 10.1002/hbm.21186.

**Von Kriegstein K, Eger E, Kleinschmidt A, Giraud AL. 2003.** Modulation of neural responses to speech by directing attention to voices or verbal content. *Cognitive Brain Research* **17(1)**:48–55 DOI 10.1016/s0926-6410(03)00079-x.

**Von Kriegstein K, Giraud AL. 2004.** Distinct functional substrates along the right superior temporal sulcus for the processing of voices. *Neuroimage* **22(2)**:948–955 DOI 10.1016/j.neuroimage.2004.02.020.

**Von Kriegstein K, Kleinschmidt A, Sterzer P, Giraud AL. 2005.** Interaction of face and voice areas during speaker recognition. *Journal of Cognitive Neuroscience* **17(3)**:367–376 DOI 10.1162/0898929053279577.

**Warren JD, Scott SK, Price CJ, Griffiths TD. 2006.** Human brain mechanisms for the early analysis of voices. *Neuroimage* **31(3)**:1389–1397 DOI 10.1016/j.neuroimage.2006.01.034.

**Watson R, Latinus M, Charest I, Crabbe F, Belin P. 2014.** People-selectivity, audiovisual integration and heteromodality in the superior temporal sulcus. *Cortex* **50**:125–136 DOI 10.1016/j.cortex.2013.07.011.

**Zaeske R, Hasan BAS, Belin P. 2017.** It doesn't matter what you say: FMRI correlates of voice learning and recognition independent of speech content. *Cortex* **94**:100–112 DOI 10.1016/j.cortex.2017.06.005.

**Zhang C, Pugh KR, Mencl WE, Molfese PJ, Frost SJ, Magnuson JS, Peng G, Wang WSY. 2016.** Functionally integrated neural processing of linguistic and talker information: an event-related fMRI and ERP study. *Neuroimage* **124**:536–549 DOI 10.1016/j.neuroimage.2015.08.064.