# Peer review of "Brain mechanism of unfamiliar and familiar voice processing: an activation likelihood estimation meta-analysis"

_PeerJ, doi:10.7717/peerj.14976_

## Round 0.1 · original submission · Minor Revisions

The manuscript is well written and the topic is interesting. Each of the reviewers have made some recommendations for improving the manuscript. These comments address improving clarity of the methodology and figure labels as well as some referencing format issues and abbreviations.

Please also respond to the question of the literature search from Reviewer 3.

·

Basic reporting

The authors conducted an ALE meta-analysis on 32 fMRI studies to investigate the brain basis of voice processing. They mainly found that unfamiliar voice processing is associated the superior temporal gyrus (STG) and familiar voice processing is associated with several temporal and frontal regions. The paper is well written and generally clear. I have three main comments (2, 4, and 5 of my bellowed points) and some other minor comments.

1. Could you please explicitly state that VIR and VII would be combined for "familiar" voice identity processing, and VIP would be for "unfamiliar" vocal identity processing in Introduction, so that readers wouldn’t expect to see the separate results of VIR and VII?

2. Is there any hypothesis of familiar and/or unfamiliar voice processing that this meta-analysis could support, against, or compare?

Experimental design

3. In line 193-194, what do you mean “the data of voice identity processing in multimodal identity processing that 194 appeared in 314 studies were considered…”? Why were these studies excluded by the original screening criteria?

4. Did you perform the comparison and conjoint analysis on results of VIR/I and VIP using different thresholds (cluster FWE, p < .01 for VIR/I; uncorrected p < .05 for VIP)? To my knowledge, it was suggested that two inputs for comparison should use the same threshold settings (e.g., the discussion in BrainMap’s forum: http://www.brainmap.org/forum/viewtopic.php?f=3&t=343&sid=59183357da9f6e04bc63d24f01ce6827).

Validity of the findings

5. How is it possible that only STG was shown significant activation in VIP individual analysis but more regions, including STG, MTG and left IFG, were revealed in P-(R+I) contrast analysis?

6. Readers like me might still be interested about the individual results of VIR and VII, though their numbers of collected studies are not sufficient for ALE analysis. Would you consider performing analysis on them separately and list the results in supplementary materials?

7. Is there any specific reason for the order of Region in Table 3 and 4?

8. Could you please add some labels in Figure 2 & 3, e.g., “ALL” in Figure 2A, “VIR/I” in Figure 2B, “VIR/I > VIP” in Figure 3A, to make readers understand the figures clearly?

9. In line 385, you stated “we found that the activation of the claustrum and caudate nucleus…” Because it is your explanation, I would suggest “one possibility is that the activation…”

10. One limitation of ALE estimation is that it doesn’t take the size of an effect into account. Thus if unfamiliar and familiar voice processing both produced significant activation in the STG, one would not know whether they have different degrees in the current meta-analysis. Would you like to discuss this point in Discussion?

Additional comments

11. Please check the formats and consistency of in-text citations. For example, line 52 & 53 should not include the author’s first name, e.g., “Pascal” or “P”. In line 210-211, for three or more authors, please list the first author’s last name and “et al.”
12. In line 418, is “P-(R+I)-P” a typo of “(R+I)-P”? In line 425, is “VIR/P” a typo of “VIP”?

·

Basic reporting

A clear, unambiguous, and professional English language is used throughout the paper.

Introduction is well written with a clear exposition of the aims of the review and of the previous investigations which have motivated it. References are well chosen and relevant.

Structure conforms to Peerj standards.
Figures are relevant, of high quality, and well labelled & described
Raw data are supplied

No suggested improvements

Experimental design

The original primary research is within the scopes of the journal.

Research questions are well defined, relevant & meaningful. It is clearly stated how the research fills an identified knowledge gap.

The investigation is rigorous and has been performed to a high technical & ethical standard.

Methods arre described with sufficient detail & information to replicate.

Validity of the findings

Impact and novelty of the findings are correctly assessed. The review is, however, considered as preliminary to a more in depth study, due to the difficulty of reaching statistical significance with data obtained up-to-date.

All underlying data have been provided; they are robust, statistically sound, & controlled

Conclusions are well stated, linked to original research question & limited to supporting results.

Additional comments

Findings are novel and should impact on the ad hoc research also in reason of the acknowledged preliminary nature of the review.
Meaningful replications, where rationale & benefit to literature is clearly stated, are therefore encouraged.
All underlying data have been provided ;they are robust, statistically sound, & controlled.

Conclusions are well stated, linked to original research question & limited to supporting results

Reviewer 3 ·

Basic reporting

First of all, I congratulate you on the work carried out. According to their study, the authors conducted an ALE meta-analysis of studies using brain mechanism of unfamiliar and familiar voice processing. Based on a literature review and screening, ten studies were included for investigating the voice identity perception, eleven studies for investigating the voice identity recognition, and eleven studies for investigating the voice identity identification. The authors concluded that the bilateral STG, MTG, and left IFG were more active in the process of unfamiliar voice identity perception than in familiar voice identity perception. The right inferior/middle frontal gyrus, right fusiform gyrus, left parahippocampal gyrus and bilateral insula were more active in the process of familiar voice recognition/identification than in unfamiliar voice identity identification

However, I have some suggestions and concerns from this study.

1. The abstract, especially in the conclusion section, needs to be rewritten because it is not very informative and just seems to rewrite the title.

2. I suggest that introduction needs more organization, especially in the second paragraph. Additionally, a suggestion is that the seven and eight paragraph seem be able to combine as one paragraph.

3. In-text styles and formats are incorrect in this article. Firstly, according to guidelines of the journal, for four or more, abbreviate with ‘first author’ et al. However, line 357 and line 372-373 are cited five authors, line 422 is cited four authors. Secondly, references by the same author in the same year should be a differentiated by letters (e.g., Smith, 2001a; Smith, 2001b), do not use incorrect format. Thirdly, the “Figure” is used an extension in line 276 and 278, but the “Fig.” is used an abbreviation in line 280. Besides, I am very confused about an abbreviation used regulation in this study. I realize that spell out the full term at its first mention, indicate its abbreviation in parenthesis and use the abbreviation from then on. However, this manuscript is not an identical standard about that. Fourthly, line and paragraph spacing is also inconsistent, such as between line 45 and 46, between line 156 and 157, among line 391-394, and so on. Finally, cited reference format is also inconsistent. Some letters in the first character use the capita (e.g., line 513, 522,532, and so on), the others use the lower case. Taken together, it is not a good writing in this manuscript. Please correct and check the format of this whole article for consistency.

Experimental design

With respect to the literature search and selection process, it is not very rigorous in this study. Although the authors have performed the search strategy and identify studies in this manuscript, being not explain 1.) whether the literature search and selection are independently done, 2.) how to extract the data and whether the data extraction have been independently conducted by two investigators, 3.) whether you met any disagreement for potentially relevant studies how to solve these problems. These questions are not mentioned and not comply with the PRISMA guidelines and the guidelines for neuroimaging meta-analyses. Therefore, this study seems to show some methodological issues.

Validity of the findings

1. This study has mentioned the study limitation. What is strength of the proposed study?

2. As you mentioned in the conclusion, bilateral STG, MTG, and left IFG were more active for processing unfamiliar voice identity perception. Moreover, the right inferior/middle frontal gyrus, right fusiform gyrus, left parahippocampal gyrus, and bilateral insula were more active for processing familiar voice recognition/identification. However, what is importance or will be able to help humanity from those results found?

Additional comments

1. Please delete a repeated abbreviation in line 121 or 123.

2. Please modify the sentences. For instance, line 60-62 (Since P. Belin, Zatorre, Lafaille, Ahad, and Pike (2000)…), line 226-227 (For…to Eickhoff et al. (2009)), and line 329-330 (…findings of Bethmann and Brechmann (2014) and Bethmann, Scheich, and Brechmann (2012)) were related to use a long author as description.

3. Please revise line 309 and 313 about ((Roswandowitz et al., 2021) added the task…) as well as (…active task of (Roswandowitz et al., 2021)).

To summary, in this proposed article, the study-selection, task-selection, format-issue, importance and strength of this study as well as the interpretation of the abstract and introduction need to be improved for quality. Consequently, based on these issues, major revision is suggested for the manuscript.

---

## Round 0.2 · accepted · Accept

All reviewers' comments have been addressed and the manuscript is now appropriate for publication.

·

Basic reporting

1. According to the APA guide, please check and adjust all "P < 0.05" to "p(italic) < .05".

Experimental design

no comment

Validity of the findings

no comment

Additional comments

I thank the authors for responding to my concerns and considering my suggestions. All my previous comments were well addressed and I'm happy to recommend the manuscript for publication.

Reviewer 3 ·

Basic reporting

The authors have modified all my concerns in the revised manuscript.

Experimental design

The authors have addressed all my concerns in the revised manuscript.

Validity of the findings

No answer given.

Additional comments

The authors have modified all my concerns in the revised manuscript.